# Patient experiences of diabetes and hypertension care during an evolving humanitarian crisis in Lebanon: A qualitative study

Ruth Willis[1,2‡*], Chaza Akik[3‡], Zeinab El-Dirani[3], Claudia Truppa[4¤],
Carla Zmeter[4], Fabrizio Fleri[4], Sigiriya Aebischer Perone[5,6], Roberta Paci[7],
Signe Frederiksen[8], Celine Abi Haidar[9], Randa S. Hamadeh[10], Fouad M. Fouad[11],
Pablo Perel[2,12], Bayard Roberts[1,2], Éimhín Ansbro[2,12]

1 Faculty of Public Health and Policy, Department of Health Services Research and Policy, London School of Hygiene and Tropical Medicine, London, United Kingdom, 2 Centre for Global Chronic Conditions, London School of Hygiene and Tropical Medicine, London, United Kingdom, 3 Faculty of Health Sciences, Center for Research on Population and Health, American University of Beirut, Beirut, Lebanon, 4 International Committee of the Red Cross, Beirut, Lebanon, 5 International Committee of the Red Cross, Geneva, Switzerland, 6 Division of Tropical and Humanitarian Medicine, Hôpitaux Universitaires de Genève, Geneva, Switzerland, 7 Danish Red Cross, Beirut, Lebanon, 8 Danish Red Cross, Copenhagen, Denmark, 9 Lebanese Red Cross, Beirut, Lebanon, 10 Primary Healthcare and Social Health Department, Ministry of Public Health, Lebanon, 11 Faculty of Health Sciences, Department of Epidemiology and Population Health, American University of Beirut, Lebanon, 12 Department of Non-communicable Disease Epidemiology, Faculty of Epidemiology and Population Health, London School of Hygiene and Tropical Medicine, London, United Kingdom

¤ Current address: CRIMEDIM—Center for Research and Training in Disaster Medicine, Humanitarian Aid and Global Health, Novara, Italy
‡ RW and CA are joint first authors.
* ruth.willis@lshtm.ac.uk

**Data Availability Statement:** The data contain potentially sensitive participant information. The participants were assured that 'the only people who

## Abstract

Humanitarian health care models increasingly incorporate care for non-communicable diseases (NCDs). Current research evidence focuses on burden of disease, service provision and access to care, and less is known about patient's experience of the continuum of care in humanitarian settings. To address this gap, this study explored experiences of displaced Syrian and vulnerable Lebanese patients receiving care for hypertension and/or diabetes at four health facilities supported by humanitarian organisations in Lebanon. We conducted in-depth, semi-structured qualitative interviews with a purposive sample of patients (n = 18) and their informal caregivers (n = 10). Data were analysed thematically using both deductive and inductive approaches. Both Syrian and Lebanese patients reported interrupted pathways of care. We identified three typologies of patient experience at the time of interview; (1) managing adequately from the patient's perspective; (2) fragile management and (3) unable to manage their condition(s) adequately, with the majority falling into typologies 2 and 3. Patients and their families recognised the importance of maintaining continuity of care and self-management, but experienced substantial challenges due to changing availability and cost of medications and services, and decreasing economic resources during a period of national crises. Family support underpinned patient's response to challenges.

will see the information that you share are the researchers involved in the study' and their data will therefore will not be made publicly available. This approach has been approved by the American University of Beirut Institutional Review Board (SBS-1019-0404) and the London School of Hygiene and Tropical Medicine Ethics Committee (16193-1). Enquiries about the data may be sent to ethics@lshtm.ac.uk.

**Funding:** This work was supported by a grant from the Novo Nordisk A/S, Global Access to Care Department to the London School of Hygiene and Tropical Medicine (EA, KB, PP, BR), as part of the Partnering for Change collaboration between the International Committee of the Red Cross (ICRC), Danish Red Cross (DRC) and Novo Nordisk. EA, PP, BR and RW received salary from the London School of Hygiene and Tropical Medicine partially supported by this grant. CA, ZED and FF received salary from American University of Beirut, contracted by the London School of Hygiene and Tropical Medicine for this work. CAH, SAP, SF, RH, CT and CZ received no specific funding for this work. The funder had no role in study design, data collection and analysis, decision to publish, or preparation of the manuscript. Novo Nordisk URL: https://www.novonordisk.com/ Partnering for Change URL: https://www.humanitarianncdaction.org/.

**Competing interests:** The authors have declared that no competing interests exist.

Navigating the changing care landscape was a significant burden for patients and their families. Interactions were identified between mental health and NCD management. This study suggests that patients experienced disrupted, non-linear pathways in maintaining care for hypertension and diabetes in a humanitarian setting, and family support networks were key in absorbing treatment burden and sustaining NCD management. Recommendations are made to reduce treatment burden for patients and their families and to support sustainable condition management.

## Introduction

Patients with noncommunicable diseases (NCDs) who are forcibly displaced in humanitarian crises face challenges of flight and displacement as well as the need for long-term management of health conditions and the attendant psychosocial impacts [1–3]. Monitoring and continuous treatment of existing NCDs is essential to prevent complications, preserve quality of life, and avoid economic burdens to the household and health system, but this is especially difficult to maintain in crisis situations [2,4]. Both the global burden of NCDs and the numbers of people forcibly displaced are increasing, and both disproportionately affect populations in low and middle income countries (LMIC), intensifying challenges for health systems [5–7]. In response, humanitarian health care models, which traditionally focused on acute, episodic care, are being adapted to include care for existing NCDs [8,9]. This mirrors the wider health system 'paradigm shift' in care for chronic conditions advocated by the World Health Organisation (WHO) in 2002 [10].

In this emerging field, attention at policy and practice levels has been on service provision, positioned from the perspectives of actors who are establishing approaches to embed NCD care in humanitarian models [1,2,7,11–14]. There has been less focus on patients' experiences of receiving NCD care during humanitarian crises, with the exception of recent research in Lebanon addressing access to primary health care services [14–16]. Our research seeks to address this gap, with a focus on the continuum of care for patients who are accessing services. We draw on the concept of continuity of care as defined by WHO: *'the extent to which a series of discrete health care events is experienced by people as coherent and interconnected over time and consistent with their health needs and preferences'* [17]. The aim of this paper is to explore the experiences of patients receiving care for hypertension and/or diabetes mellitus (HTN/DM) at four health facilities supported by humanitarian organisations in Lebanon during 2020–21. Related research from the health system perspective will be published separately.

This study was conducted within a wider research project exploring the continuum of care for hypertension and diabetes in LMIC humanitarian settings, under a partnership between academic, national humanitarian and governmental partners and 'Partnering for Change' (P4C) [18], including the Lebanese Red Cross, the Danish Red Cross (DRC) and the International Committee of the Red Cross. The partnership identified hypertension and diabetes as focal conditions due to their high disease burden in the study context [19,20] and globally [21], frequent co-existence and amenability to a primary level public health approach for management and prevention of complications.

Lebanon's health system has been challenged by a series of crises over several decades. The protracted civil war (1975–1990), and subsequent political and economic instability have threatened the Ministry of Public Health (MoPH)'s governance capacity, leading to rapid, unregulated expansion of the private sector and nongovernmental organisations (NGOs) [22–

26]. The system developed along sectarian lines, with multiple primary care providers and a reliance on hospital-based and privatised healthcare, leading to geographical and income-based disparities in access [27,28]. The protracted Syrian crisis, beginning in 2011, resulted in Lebanon hosting the largest number of refugees per capita globally in 2019 [29]. With international support, the Lebanese public health system has accommodated over 1 million Syrian refugees [30].

Current models of care for HTN/DM sit within this pluralistic health system, with primary health services provided mostly by local entities (governmental or NGOs), which may be supported by international actors, and privatised hospital services. MoPH support to bolster primary care provision has included strengthened procurement, training and guideline development, including for NCDs [31], introduction of an accreditation and national monitoring system [32] and an essential health care benefit package [33]. While NCD service coverage for Syrian refugees has reportedly been high (83%), access for Syrians and vulnerable Lebanese has been limited by cost, availability and legal issues, and services have been fragmented, of variable quality, and poorly coordinated [16,27,34–36].

The country has been further destabilized by an economic and political crisis (2019), the Beirut port explosion (2020) and the COVID-19 pandemic [26,37–39]. The economic crisis resulted in inflation, fuel and electricity shortages, MoPH funding cuts, reduced ability to import essential drugs and equipment, staff migration, curtailment of services at hospitals and primary level facilities and stock outs of essential medicines and vaccines [38]. The Beirut port explosion physically destroyed supplies and infrastructure and the pandemic further diverted finances, services and supplies from routine care [27]. The population has been further impoverished and the burden on the public health system has increased.

## Methods

### Study setting

The study was set in four health facilities providing NCD services, supported by P4C partners and serving urban catchment areas with substantial refugee populations (estimated 30–45% of total catchment population) in North Lebanon and Mount Lebanon governorates. Two were primary-level dispensaries, one a primary health care centre, and one a division of a governmental tertiary hospital providing emergency inpatient care. All provided low cost or free-of-charge health services to Syrian refugee and vulnerable Lebanese patients with previously diagnosed hypertension or diabetes.

### Data collection and management

We conducted in depth, semi-structured qualitative interviews with patients (n = 18) and informal caregivers (i.e. family members/neighbours/friends who support patients with their NCD condition) (n = 10). Sample size was guided by experience from previous research with parents and caregivers and by project resources. Patients aged 18 years or over, diagnosed with HTN/DM at least 5 years ago, and receiving care at one of the four study health facilities were eligible for interview. Patients were purposively sampled to represent a range of demographic (gender, age, nationality) and condition-related (hypertension and/or diabetes, with/without complications, such as previous heart attack, stroke, diabetes-related amputation or visual impairment) groups, and then randomly selected from within these groups. Sampling criteria were identified by the humanitarian study partners in consultation with facility management. The sample was broadly representative of the HTN/DM patient population at each facility, weighted to ensure representation of those with known complications, in order to include experiences at multiple health system levels. Following COVID-19 social distancing guidance,

patients were contacted by telephone for invitation to participate, using an oral recruitment script, and subsequently for telephone interview from a private university office. Potential participants were provided with the study information sheet and consent form electronically in advance, and these were read out over the telephone, time given for questions, and verbal consent recorded in a consent log. Participants were assured that participation in the study would not affect the current care received and interviews were arranged at a time convenient for participants. At each patient interview, patients were asked whether someone assisted in caring for them, and, if so, permission was sought to contact this person for study recruitment. Participants were reimbursed for their time with a phone credit or cash vouchers at the patient's next visit to the health facility.

Patient and carer topic guides (S1 Text), informed by conceptual frameworks for care for chronic conditions [9,40], were developed in English (CA, EA, NER, RW), reviewed by MoPH and humanitarian study partners, translated into Arabic (CA, ZED) and piloted (CA, ZED). Interviews lasting 60–90 minutes were conducted between September 2020—February 2021 in Arabic by a female research assistant (ZED, MPH) fluent in Arabic and English with training and experience in conducting qualitative research with vulnerable populations. Regular debriefings with the AUB/LSHTM research team were conducted during data collection, and fortnightly update meetings were held with study partners to discuss emerging findings. Interviews were audio recorded, transcribed verbatim in Arabic by the research team and translated into English language by professional translators (NN, BT). In order to ensure consistency and inter-reliability in translated terms, the research team developed a glossary of key terms in Arabic and their translation to English. To ensure accuracy, translated transcripts were checked against the original version by the interviewer.

## Data analysis

A coding framework for thematic analysis was developed using a combination of inductive and deductive approaches, guided initially by the overarching project conceptual framework [9] which focused on health system factors. Four researchers (CA, EA, ZED, RW) independently coded a subset of transcripts. Differences were reviewed and resolved, and the coding framework was iteratively adapted in response to emerging themes. Comparisons were made between participant sub-groups and deviant cases were explored. Our analysis involved several stages undertaken for different purposes: first, producing preliminary findings to share at a stakeholder workshop and inform humanitarian partner programmatic decisions, then focusing in greater depth on themes prioritised by stakeholders, then further in-depth analysis of specific cases or case pairings, including deviant cases, some through open coding. This paper presents a subset of the results, which cut across different sections and levels of the initial coding framework. The typologies and themes reported were primarily identified inductively. NVivo for Windows software was used to support analysis.

## Patient and public involvement

Patients were involved as research subjects, with the study aiming to understand their experiences of care in order to identify ways in which services can be adapted to better meet their needs. Patients/the public were not involved in the design or conduct of the study, or in dissemination of results.

## Inclusivity in global research

Additional information regarding the ethical, cultural, and scientific considerations specific to inclusivity in global research is included in S2 Text.

### Ethical approval

The study was approved by the MoPH Lebanon following submission of a letter of intent from study partners, and an MoPH focal point was appointed to the study team. Approval was received from AUB IRB (SBS-2019-0404) and LSHTM Ethics Committee (16193–1), including an amendment to data collection methods due to COVID-19 social distancing requirements. The study conformed to the principles embodied in the Declaration of Helsinki.

## Results

We interviewed 18 patients and 10 caregivers. Eleven patients initially identified as eligible did not participate (three did not meet criteria, seven were not contactable, one declined). Patients were aged between 29–73 years, with two patients aged under 40 years. Equal numbers (n = 9) were male and female. Ten patients were Syrian and eight were Lebanese. The majority of patients (n = 12) were married, and three were single, two divorced and one widowed. Four patients had been diagnosed with hypertension, eight with diabetes, and six with both hypertension and diabetes. One patient had been diagnosed three years prior to being interviewed; the remaining 17 had been diagnosed 5–20 years previously. Ten patients reported having experienced complications (two patients with hypertension; three comorbid patients, five patients with diabetes).

Caregivers were identified by twelve patients, and ten were contactable and willing to participate in interviews. Caregivers were aged 27–65 years, with five aged under 40 years. Equal numbers were male and female. Seven were Syrian and three were Lebanese. All were family members of patients, including spouses (n = 4), children (n = 3), one parent, sibling and cousin. Two caregivers reported having hypertension, and one of these had experienced complications.

One-third of respondents reported being illiterate and one-third had primary school education only. Two-thirds of participants were unemployed and 2 of 28 participants were in formal employment. All participants reported worsening financial situations due to the economic crisis in Lebanon and COVID-19 pandemic.

Patients reported on their experience of managing their condition from first symptoms to current care. As all patients had been diagnosed over three years previously, they described experiences of care both at the facility through which they were recruited and at other facilities they had attended previously (and, in many cases, simultaneously). Here we summarise findings about patients' experiences across three key points in the health care pathway for diabetes and hypertension: at onset of symptoms and diagnosis, during daily management, and at development and care for complications (if any). We describe three typologies of patient experience identified in our data across both conditions, presented through three case studies of individual respondents exemplifying those experiences. We then examine challenges for patients and carers and the strategies which they draw on to manage these challenges.

### Care pathway

**Onset of symptoms and diagnosis.** Patient's experiences at diagnosis and conceptualisations of disease causation can influence their trajectory of condition management, hence we sought to understand why and how patients had initially sought care. Preceding diagnosis, the majority of patients had experienced symptoms which interfered with their daily lives and prompted them to seek medical care. Some reported seeking care independently, and others being urged to do so by family or friends, whose involvement ranged from making suggestions to arranging medical appointments or performing blood glucose testing. For example, a male Lebanese patient reported being diagnosed after a friend recognised his symptoms and tested him with his glucometer: *'I told him that I do not have diabetes. He told me "yes, you do"'*

[Participant H, DM]. One Syrian patient had been diagnosed through free mobile diabetes screening in a camp setting in Lebanon, which they attended after feeling unwell. Otherwise, no patients reported diagnosis via community based programmes.

When describing early symptoms, a number of respondents, both Syrian and Lebanese, spontaneously positioned disease onset in terms of what they saw as causal events in the patient's life, particularly for diabetes. Ten cases of diabetes were attributed to experiencing extreme fear, shock or sadness, for example after the death of a family member, or other specific traumatic events, such as their children being attacked or themselves having been held hostage during armed conflict. Association between traumatic experiences and disease onset was described particularly by Syrian respondents, including a patient who attributed their diabetes diagnosis to sadness caused by *'the disasters and the news in Syria'* (Participant K, DM). Several respondents noted that, while diabetes could be hereditary, this was not so in their case, for example *'I have diabetes because of a scare, it's not hereditary in my family'* (Participant D, DM/HTN, Syrian). Fewer participants gave spontaneous explanations of hypertension disease onset. Several identified family members with hypertension, noting that the condition could be hereditary, and two patients related it to psychological distress, for example '*According to my information, I do not know if it is right or wrong. Hypertension results from sadness or anger'* (Participant M, HTN, Syrian).

After first experiencing symptoms, most patients had consulted a range of providers for both diagnosis and care, including pharmacists, physicians in both private and public clinic settings, and hospital emergency departments. An exception to this was a female Lebanese patient whose blood glucose reading was high when tested by her diabetic mother and at a pharmacy, but who had not consulted a physician until 10 years later when she experienced leg pain while accompanying her mother to a hospital consultation. She explained that she *'did not like to follow up with the doctor'* and '*thought it was high because of sadness*' caused by the '*shock*' of her brother's sudden death (Participant S, DM).

**Daily management.** Respondents primarily described their management of hypertension and diabetes in terms of the worsening availability and affordability of different elements of care, foregrounding their challenging daily experiences of managing chronic conditions with increasingly constrained resources. All patients had substantial experience of living with one or both conditions, and had sought care across multiple facilities within Lebanon, or in Syria and Lebanon. For both conditions, patients recognised the need for regular clinical monitoring and daily medication, both of which they found increasingly challenging to access as the situation in Lebanon changed. Both Lebanese and Syrian patients reported recent reductions in their income, as employment opportunities reduced due to the financial crisis and pandemic, in parallel with reduced medication availability and increased costs. For example, a Syrian refugee with diabetes and hypertension reported that subsidised medication was no longer reliably available at the dispensary she attended, and she could not afford to purchase it privately: '*She tells me to get them from elsewhere, and I really don't have the money. Both my husband and I need medicine. That's why we always have a shortage in medicine. I'm not taking all my medicine...[...]...I don't have the money to buy them'* (Patient C). These changes resulted in widely reported gaps in both clinical monitoring and medication use, and in increasing time being spent by patients and their families searching for medicines, as discussed further below.

The majority of patients also discussed basic dietary management, including weight control (both conditions), salt reduction (hypertension) and avoiding sugar and carbohydrates (diabetes). Some participants reported excluding proscribed foods: '*I can control myself and not eat sugar'* (Patient B, DM/HTN, Lebanese), while others reduced portion sizes '*I eat a small portion'* (Patient G, DM, Syrian), and some recognised the need to make changes but were unable to '*This is who I am. I cannot resist'* (Patient H, DM, Lebanese). Dietary management was

generally seen as of secondary importance to medical care, and reported to be sometimes impractical due to food cost or availability. Respondents also reported patients being advised to keep calm and avoid stressful feelings and situations in order to help manage their condition, for example a diabetic patient explained '*I try to avoid being with annoying people. I try to avoid being mad at my children*' (Patient G, DM, Syrian).

Overall, the pattern of active management practices described by the majority of patients at the time of interview was interrupted rather than continuous. This largely reflected gaps in affordability and availability, explored further below, rather than gaps in patients' understanding of the need for continuous management.

**Development and care for complications.** Patients' experiences of condition management prior to complications included (i) not being diagnosed with their condition prior to experiencing complications (e.g., myocardial infarction, diabetic ketoacidosis); (ii) being diagnosed and not adhering consistently to dietary guidance/medication because they did not consider it important or a priority, particularly for diabetes; (iii) being diagnosed and not adhering to medication/engaging in monitoring because they could not afford to. Several patients in category (ii) reported having changed their approach after experiencing complications, for example a Syrian male who had not followed dietary guidance and been hospitalised had severe, life-threatening complications, undergone minor amputation and been assessed for major amputation explained '*It is not a joke. I neglected it, but when I had a problem with my leg, everything has changed.*' (Patient Q, DM).

Some patients had received care for complications in a variety of previous settings in Syria and Lebanon, and recently in a study partner-supported hospital setting. Similarly to daily management, patients' experiences of care for complications were framed primarily in terms of affordability, although reported costs were vastly different, with inpatient hospital costs quoted to patients proportionately up to 450 times greater than outpatient consultation or medication costs (reported between US$2500–4500, exchange rate 1500 LBP = US$1). Costs were heavily subsidised for eligible patients in emergency situations, but eligibility status was understood by patients as contested and temporary, with patients describing moving in and out of 'emergency' classification over time. Patients expressed frustration that despite having severe symptoms of complications which severely affected their daily lives and gradually worsened, interventions could only be subsidised when they became very ill, for example '*they want someone who is dying so he can be considered as an emergency case*' (Patient M, HTN, Syrian). Once established, eligibility remained dependent on other factors beyond the patient's control and could be revoked. A diabetic patient's caregiver explained that approval granted from a partner organisation for surgery costs had expired after the doctor postponed the planned surgery date '*the doctor said he will do it next Monday...[...]...they told us if we adjourn the surgery until Monday they won't be able to cover the expenses*' [Caregiver R2]. Here the procedure may have been approved as immediately lifesaving care, which could not be postponed. These examples illustrate the fragility of continuity of care for patients who are accessing services.

## Typologies of patient experience

Describing experiences at different points in a clinical 'care pathway' has a tacit framing of linearity; entering a defined pathway and moving along it over time under the supervision of a consistent system of care provision. However, we found that although the majority of patients recognised the principle of continuous management, few described a continuous pattern of care, monitoring or regular attendance at a single provider. Care seeking was prompted by needing medication refills or feeling unwell, and was related to severity of symptoms, financial resources and competing responsibilities and priorities.

We describe three typologies of current patient experience, across both conditions, that emerged from our data. These cut across gender and nationality sub-groups. We illustrate these typologies through case-study examples of three individual respondents whose experiences exemplify the respective typology, highlighting the ways in which an individual's situation influenced their capacity to engage with and navigate care systems.

**Typology 1: Managing adequately from the patient's perspective.** The patient is able to obtain medications to manage their condition, monitor their condition, either at home or by attending a facility, and follow dietary and exercise guidance. They have mild or no symptoms, which do not interfere in their daily life. Very few participants were within this typology at the time of interview. A number of typology 2 and 3 patients reported previously having been in this situation, prior to changes in their individual/wider economic and social situations and/or to developing complications.

Patient A is a married Lebanese male with elementary level education, in stable employment, who lives with his family. Over ten years ago, in his mid 40's, he was diagnosed with hypertension by a pharmacist after feeling pressure in his head. He sought care in the private sector from a cardiologist, who also diagnosed diabetes, after which he sought care from an endocrinologist. He took regular medication for both conditions and his condition stabilised. He received verbal guidance and a brochure about diet and '*started to pay attention*' to what he ate, and also '*followed up on the internet*', explaining that, while he initially had '*little knowledge*' about the conditions, '*I now try to take care of myself as much as I can*'. He reported exercising daily and prioritising regular medical care, explaining '*the most important thing to me is to see the doctor, I mean to have a check-up and show him the tests I do every 3 months*'. After previously purchasing medicine privately, he started obtaining medicines from a study partner-supported facility '*because the financial situation isn't that good*'. Initially medications for both conditions were available at subsidised cost, then hypertension medication became unavailable and diabetes medication was restricted by half. At the time of interview, he obtained medication from two subsidised facilities supported by different organisations, and, if it was not available in either, he purchased it privately.

**Typology 2: Fragile management.** The patient is aware of the main measures needed to manage their condition, but cannot always follow these fully, due to lack of resources, and may not fully understand their condition(s). They may have complications, and have symptoms which sometimes interfere in daily life. The majority of patients were in this typology, some of whom described past experiences aligned with typology 1.

Patient N is a married, uneducated female Syrian refugee aged 50–54 years who was diagnosed with hypertension and then diabetes over 10 years ago in Syria. Four years ago she travelled from Lebanon to Syria to have a cardiac catheterisation due to a blocked artery, which was unaffordable for her in Lebanon. She experiences a '*constant headache*' which she associates with her conditions, and sometimes needs to rest in bed for 5–10 day periods. If this happens, her family support her by dividing up her housework between them, getting her medicine and looking after her: '*my husband, my children, my girls. They all help me*'. Patient N has received dietary advice and been advised to exercise, which she does by walking '*as much as I can*', although her adult son discourages her from exercising '*because of her health state*'(Respondent N2). She is aware that she should always take medication '*pills for hypertension, pills for diabetes and anticoagulant pills. I should never stop taking them*'. Her son obtains her medication and explained that although '*one can barely afford food*', he ensures that she '*takes her medication daily*', although all medications are not always available. Before the COVID-19 pandemic, her son arranged for her brother-in-law or a taxi driver to travel to Syria to purchase medication, because this cost less than purchasing in Lebanon. During the pandemic, travel was stopped and her son purchased her medication in Lebanon at higher

cost. She has visited '*over 10 doctors*' in Lebanon on the advice of friends and family members '*because I don't get better*'. Her adult son explained that they have visited many doctors trying to find '*a medicine and a diet to follow*' in order to stabilise her condition.

**Typology 3: Patient feels unable to manage their condition(s) adequately.** Patients in this category may be aware of the steps needed to manage their condition, but are unable to do so, due to lack of resources/ support, and may not fully understand their current condition(s). They may experience complications and their daily life is severely affected by their symptoms. A substantial minority of patients were in this typology, and most had complications.

Patient S is a divorced, unemployed Lebanese female with secondary school education, aged 55–59 years who lives alone. She received high blood glucose test results 15 years previously, attributed this to 'sadness' (described above) and started to receive care 10 years later from a private physician when she was experiencing leg pain. She saw several private physicians and had surgical procedures related to complications, partly funded by the MoPH. She reported that the co-payments required were unaffordable for her. During one procedure, she experienced further complications, and later a leg ulcer which did not improve with antibiotics. During this period she had problems reducing her blood sugar level and was prescribed insulin. She was referred between doctors, including a cardiologist and endocrinologist, whom she felt '*only cared about the fees*', and the wound worsened. She explained '*in the end, I lost hope from the different doctors in the different clinics; I was going to the clinics, paying the fees, and leaving. The doctors were not responsive. They used to tell me that, 'We need money. There are the Corona pandemic and the financial crisis*'. She later received care with the support of a partner organisation, but after their involvement was disrupted by COVID-19 quarantine, her condition worsened and her leg was amputated below the knee. She valued the involvement of a doctor from the partner organisation who followed up with her by telephone and arranged hospital referral when she experienced symptoms in her remaining leg, but reported, '*now they asked for another surgery and I do not know what to do*'. At the time of interview, she explained that she was treating her swollen leg with ice. Her medications were '*expensive*' and after her amputation she had sold belongings to help pay for them. Her daughter and son-in-law live abroad and help to provide for her. She explained '*if my daughter calls me now, I laugh and joke with her, and I do not let her know that I am sad*'.

## Maintaining continuity of DM/HTN care: Key factors identified by patients and caregivers

Across the three typologies of management, four themes were identified which iteratively influenced maintenance of continuity of care: affordability and availability of medication and care, navigation of the health care system, family support, and mental health.

**Affordability and availability of medication, laboratory testing, and routine care.** The worsening financial situation in Lebanon, exacerbated by the COVID-19 pandemic, affected patients' ability to afford even minimal health care costs, with implications for continuity of care. As highlighted in the typologies above, being able to afford care was a key factor in patients' ability to manage their condition. Although some providers sought to provide free or affordable care with free-of-charge consultations and subsidised medications, patients reported expensive or unaffordable out-of-pocket costs including costs of monitoring tests, registration fees, prescription renewal and medications: '*you can't keep up with paying for doctors, laboratories, and medicine*' (Patient B, DM/HTN, Lebanese). A caregiver for two adult diabetic siblings, who receive sporadic financial support for insulin purchase, explained '*they both need to do the HBA1C to know their blood glucose levels. But I can't afford the tests, I can barely afford the insulin shots*' (Caregiver R2, Syrian). Several patients owned glucometers for diabetes self-monitoring, but could not afford to buy replacement lancets/strips.

As well as the ability of patients to pay for care decreasing, medicine availability had become variable, increasing costs where patients needed to purchase privately from pharmacies or move between providers, for example: '*medication. . . it became expensive, especially during this period. After all, medications are unavailable*' (Patient C, DM/HTN, Syrian). Subsidised medications were important for patients to maintain adherence, and when these were unavailable, some patients could not afford to purchase privately, resulting in gaps without medication: '*I did not find them at the [x organisation], and I cannot afford to buy them from the pharmacy*' (Caregiver I2, DM/HTN, Lebanese).

Where challenges of affordability and availability could not be overcome, elements of care were reduced, substituted or removed. In some cases, this happened in a managed way, with the patient or caregiver taking pragmatic decisions to maintain key elements of care as far as possible, for example a caregiver described his mother rationing her medication '*She takes one pill per day. Instead of one pill in the morning and another one in the evening, she says that way, it can last longer.*' (Caregiver I2, DM/HTN). Where medication gaps were unavoidable, non-pharmaceutical substitutes were reported, for example '*I go to the [x organisation] and they tell me they don't have it. I'm currently eating garlic with yogurt*' (Patient J, HTN, Syrian).

Where regular monitoring had become unaffordable, patients prioritised purchasing medication with the minimum consultation required in order to maintain treatment. For patients who felt unable to manage their condition (Typology 3), reductions in care were greater and more haphazard. Some patients were not taking any medication at the time of interview, or not accessing care for complications, because they could not find affordable ways to do so. Some of these patients described feelings of hopelessness or needing to 'ignore' complications which impaired their mobility, because they could not identify ways to access care and did not have support to do so, or their caregivers were also unable to find solutions.

Availability of medicines and services from different providers was discussed in terms of quality of care by some patients. There was a clear sense that overall provision of care had been more reliable prior to recent events (the national economic and political crisis, the Beirut port explosion and the COVID-19 pandemic), and that the increasing numbers of patients seeking support at public facilities led to greater pressure on the system, resulting in long wait times and short consultations. This theme is explored further in related research from the health system perspective which will be published separately.

**Navigating the health care system.** Patients and caregivers reported needing to identify and visit multiple health care providers to piece together the necessary elements of their care including consultations, monitoring tests and medications. For participants, this process was a significant and time-consuming aspect of condition management.

Most patients visited different providers for consultations and testing. For example, a hypertensive patient's wife (L2) explained, '*He goes to a private laboratory for some tests because the tests are unavailable at the dispensary. And if he wants to do an ultrasound, he has to go to a private centre*'. In some cases, patients reported being referred to specific private providers, in others, patients/caregivers needed to identify providers themselves, or searched for cheaper options than those suggested. Unusually, one patient reported having received care at a subsidised facility that offered testing: '*they have an ECG. They also have a laboratory for medical tests for free or a symbolic fee. It costs 3000 [~2 USD]. I am honestly satisfied. . .*' (Patient K, Syrian, DM/HTN). However, his medication was unavailable at this facility, necessitating visits to other facilities to try to obtain it. As well as identifying providers, arranging and paying for testing, patients were also responsible for transferring their information between providers, for example, '*She would tell me to do certain tests and to bring her the results*' (Patient B, DM/HTN, Lebanese). This patient responsibility for co-ordination was reported both within and between facilities. For patients seeking in-patient care for complications, participants reported

visiting multiple organisations in search of subsidised support, and negotiating gaps between coverage from different organisations.

Gaps in availability of medications and non-pharmaceutical products also generated additional tasks for patients and their caregivers when they needed to search for alternative sources. This was widely reported for medication, for example, a Lebanese patient whose family was searching for a hypertensive medication at different pharmacies explained '*My husband and my children have been looking for it for three days*' (Patient F). Patients also reported having been asked to provide single use equipment for medical procedures, and in one case an artificial artery: '*He told me, "you need an artificial artery. You should pay because it is unavailable at the hospital"*' (Patient S, DM, Lebanese). In this case, the patient was given a product code, contacted a supplier and visited them to purchase and collect the artery, and delivered it to the hospital.

Navigation therefore involved patients and caregivers collecting information about where key elements of care were available and physically visiting different places to obtain care, and also in taking responsibility for transferring information and resources between different care providers.

**Importance and scope of family support.**   When patients faced challenges in maintaining continuity of care (Typologies 2 & 3), family support underpinned their responses. This included financial support to pay for care, practical and emotional support, from both immediate and extended family. Both Lebanese and Syrian families drew on their wider social networks to identify and access alternative sources of medications, for example, sent from overseas, and share information about subsidised care. Patients who felt unable to adequately manage their condition (Typology 3) tended to have more limited immediate family support networks, including single, divorced and widowed patients or caregivers, and the majority were Syrian. For some caregivers, there were significant personal implications of providing support to patients, for example stopping work for periods to assist patients who were immobile to attend daily wound debridement. Caregivers also reported adapting household member's behaviour to help support patient's mental health and avoid causing worry or anger, for example: '*I tell my kids to be quiet and not say anything next to your father. We accept anything so we can support him morally because he is sick*' (Caregiver T2). As discussed, provision of support could become overwhelming and caregivers exhausted.

**Mental health.**   Both patients and their caregivers described challenges related to stress and control of emotions resulting from difficulties in maintaining continuity of care, in tension with other factors. These were reported by both Syrian and Lebanese participants, but traumatic experiences related to conflict and displacement were more frequently described by Syrian participants.

Some patients were advised by health care providers to avoid feelings of stress and anger in order to help control their conditions, and reported that they found this challenging, given the difficulties they faced in affording health care and daily living costs. This was particularly pronounced for younger patients who had responsibility for dependent children. For example, a 35–39 year old unemployed Syrian male with cardiovascular disease complications explained, '*the first thing that the doctor recommended to me is to avoid stress. Stress is the main factor for hypertension*', but he found this advice difficult to follow because he could not afford to support his family or pay for his own health care '*when you have a bad financial situation and have six kids, you have to meet their needs. But you cannot do that, so you become sad*' (Patient M). An unexpected barrier to managing feelings of stress or anger was identified by a caregiver who explained that her relative '*used to smoke a cigarette whenever he felt angry*' (Caregiver T2, Syrian) which helped him to relax, but '*because he stopped smoking*' following medical advice, he was now less able to alleviate angry feelings.

Caregivers' accounts also showed wider mental health impacts on the family of providing long-term support for a patient, particularly for patients among the 'fragile management' typology (2). Their caregivers were frequently concerned about being unable to afford routine care or find scarce medications, or being unable to help their family member in case of deterioration in their condition, for example an adult son supporting his mother explained *'I am afraid of something that might happen to her, and I cannot do anything for her'* (Caregiver I2, DM/HTN, Lebanese). The adult son of a 60–64 year-old male patient with diabetes and hypertension who could not afford to consistently purchase medications explained *'If someone is sick, the whole family, they break down, everyone gets sick, their mental health deteriorates. . . The diseased person is sick and the family gets mentally tired'* (Caregiver J2, Syrian).

## Discussion

Focusing on patients from vulnerable Lebanese and Syrian refugee populations already accessing care for hypertension/diabetes, this study has highlighted the challenges experienced by patients and caregivers in maintaining continuity of care in a humanitarian setting. Based on participants' characterisation of their situation, we identified three typologies: (1) adequate management; (2) fragile management; and (3) feeling unable to manage their condition adequately. We found that patients' positions within these typologies were not static, and identified key factors influencing movement between typologies.

The study was designed to allow comparison between experiences of vulnerable Lebanese and Syrian refugee patients, and between genders, as noted elsewhere [15,16,41]. The results found no clear differences in participants' experiences of the continuum of hypertension/diabetes care by nationality or gender in the study sites. This may be due to the extreme economic changes in Lebanon worsening the situation of all vulnerable patients, reducing their capacity to engage with care and compressing the range of experiences reported. Along a continuum, Lebanese patients tended to be in typologies 1 and 2, and Syrian patients typologies 2 and 3, but with substantial overlap.

Across both patient groups our study highlighted interactions between mental health and NCD management, resonating with regional and international evidence about mental health impacts of conflict and trauma and the synergy between mental and physical chronic conditions [42–46]. Interactions were consistently identified by respondents in relation to disease causation, feedback loops of managing stress in conjunction with concern about its impact on disease outcomes, and mental health impacts of sustaining long term condition management in an unstable context.

Below we draw on the data presented and learning from other settings to explore two wider themes underpinning these findings.

### Non-linearity and disruption of patient pathway

We found that clinical care provision was not experienced by patients as a reliable, consistently available system, with linear, consecutive stages, but as a changing landscape of elements and gaps pieced together by patients and their family support network. Participants' accounts of experiences of care focused on their day-to-day ability to manage the patients' condition in a rapidly changing context, and the continual disruption and reconfiguration of care which this involved. The pluralistic system elements are a combination of the pre-existing system and support provided by a range of humanitarian actors in specific project cycles [47–49], with fragmentation compounded by recent crises diverting or destroying resources. This non-linearity resonates with findings of a recent review of pathways for hypertension care and control in LMICs, which suggests that patient pathways 'are best characterised as continual cycles of entry and re-entry into the system'[50].

In exploring the continuum of care for patients registered with the study health facilities, challenges in maintaining continuity were anticipated to emerge particularly at movement between different levels of care. However, we found that patients also experienced substantial challenges in maintaining access to care within each level, because i) some necessary components such as laboratory testing were not routinely provided at all sites, and requirements were not consistent between providers (e.g. specific tests and test intervals); ii) provision changed, for example medications became unavailable, or iii) the patient's individual and wider contextual situation changed, affecting their ability to access available care, for example, they could no longer afford to pay subsidised nominal fees. Respondents therefore described disrupted engagement with care within each level of the clinical pathway, as well as at points of transition. The importance of logistical factors, including fluctuating medication availability, for retention of patients in care has been highlighted across LMIC settings including Bangladesh, Cambodia, Kenya and Uganda and in humanitarian settings [8,51–54].

## Responsibility for maintaining continuity of care, patient capacity and burden of treatment

In patients' experience of disrupted pathways of care, moving between healthcare providers and seeking to fill gaps in care, responsibility for maintaining continuity of care was held by patients and their family support networks. This represents a widening of responsibility from a patient role of adhering to medical guidance, monitoring and taking medication, to also actively seeking out and maintaining access to necessary elements of care. The delegation of 'work' from a health system to patients with chronic conditions and their social networks has been described in relation to the concept of 'burden of treatment', which recognises the workload of healthcare and its impact on patient functioning and well-being [55,56]. For participants in this study, a substantial additional 'workload' was generated by the changing availability and cost of key elements of care, in parallel with patients' decreasing ability to afford them.

The concept of a balance between the demands or workload placed on patients and their capacity to address these demands is the central mechanism of a conceptual model of patient complexity, developed to explain 'how clinical and social factors accumulate and interact to complicate patient care' [57]. In this model, a 'workload of demands' includes all the everyday tasks which an individual needs to undertake and accommodate, related to managing their condition and to all other aspects of life. Capacity to address demands encompasses abilities and resources, including physical functioning, literacy, social support, and economic resources. An imbalance between the two, where workload exceeds capacity, is proposed as 'the primary driver of disruptions in care, self-care, and outcomes'[57]. People in crisis situations are likely to have a greater workload of demands, stemming from instability at individual level and system level, and less capacity, with fewer available economic resources and more demands on social support networks. As shown in our Typology 3 case study, care is more complex and expensive, and morbidity and burden of treatment are much greater for the patient and their family if the disease is not detected early and well controlled.

Applying this framing to our study, immediately prior to and during the study period, patients were experiencing a simultaneous increase in 'workload of demands', related to both their burden of treatment and other changing demands such as finding employment and food, and a decrease in capacity, as economic resources were depleted and social support networks put under increasing pressure. This imbalance reduced respondents' ability to maintain management, and patients moved between typology 1–2 and 2–3. Where patients had greater capacity to address their workload of demands, for example through proactive family support

networks, they were more likely to maintain fragile management of their condition, and avoid moving to a situation where they felt unable to adequately manage. This conceptual framing highlights the importance of patient capacity, and the expanded treatment burden which arises for patients and their support networks in being delegated responsibility for maintaining continuity of care. In other humanitarian and LMIC settings, models of care have been developed to help reduce this burden through proactive engagement of family or community treatment supporters, some building on experiences from HIV treatment. These include integrated HIV/NCD medicine adherence clubs [58], UNRWA's Family Health Team approach which addresses individuals within their family units [59] and diabetes treatment support in DRC [60].

## Strengths and limitations

Strengths of our study include adding to the limited literature on patient perspectives and including caregivers' perspectives, and applying burden of treatment framing to understand experiences of obtaining care for a chronic condition in a fragmented system. The scope of the study population was limited by the need to conduct telephone interviews, excluding patients without access to a telephone who may experience greater challenges in maintaining care. By focusing on experiences of continuity of care among patients receiving care, those who were not diagnosed or not accessing services were outside the scope of the study.

## Conclusion

Our study suggests that patients experienced disrupted, non-linear pathways in maintaining care for hypertension and diabetes, and family support networks were key in absorbing treatment burden and sustaining condition management.

We have six main recommendations, summarised in Fig 1. First, recognise and act to reduce treatment burden for patients and caregivers where possible. This may include (i)

Six recommendations to support sustainable condition management and reduce treatment burden for patients and their families:

1) Recognise and act to reduce treatment burden for patients and caregivers where possible.
2) Determine, recognise and support the role of social support networks in assisting patients to maintain continuity of care.
3) Consider where NCD prevention and early diagnosis could be incorporated into existing programmes during protracted humanitarian crises.
4) Integrate mental health support.
5) Engage in efforts to develop and apply shared, evidence based clinical guidelines within humanitarian settings.
6) Work with MoPH to facilitate (i) strengthening of existing primary care systems, referral pathways and supply chains;  (ii) coordination of the humanitarian sector; and (iii) regulation and oversight of the private sector.

**Fig 1. Summary: Six recommendations to support sustainable condition management and reduce treatment burden.**

raising awareness among staff through incorporating concepts of patient burden and capacity into existing patient-centred care training and clinical guidelines [61] including training clinicians to recognise the stress for patients of crossing any care boundary within or between health care providers [17,62]; (ii) identifying aspects of the model of care which can be reinforced, supported or adjusted to reduce treatment burden, for example through introduction of an information system with a shared platform and standardised treatment guidelines; iii) supporting navigation of the system, for example through introducing or reinforcing a designated social worker/key person role.

Second, determine and recognise the role of social support networks in assisting patients to maintain continuity of care, and identify context appropriate, culturally-grounded ways to actively support caregivers [63], or to strengthen support for patients who do not have adequate family support. Actively supporting caregivers may include recognising the role of family support networks and facilitating their engagement with health care providers, for example, through invitation to attend appointments, targeting caregivers in community outreach, and offering mental health and psychosocial support to both patients and caregivers. Third, consider where NCD prevention and early diagnosis could be incorporated into existing programmes during protracted humanitarian crises, to help avoid a much greater, potentially unmanageable burden of treatment for people, including due to development of complications. Fourth, integrate mental health support through strengthening staff knowledge on mental health and co-morbidity and impact on managing diabetes/hypertension; implementing routine mental health screening; integrating mental health/DM/HTN services and support.

Fifth, engage in efforts to develop and apply shared, evidence based clinical guidelines within humanitarian settings, informed by existing tools to support continuity and coordination of care [17], to maximise continuity and provision of rationalised, cost-effective care. Sixth, work with MoPH to facilitate (i) strengthening of existing primary care systems, referral pathways and supply chains; (ii) coordination of the humanitarian sector; and (iii) regulation and oversight of the private sector.

## Supporting information

**S1 Checklist.**
(DOCX)

**S2 Checklist.**
(PDF)

**S1 Text. Semi-structured patient or carer interview guide: English language version.**
(DOCX)

**S2 Text. Inclusivity in global research questionnaire.**
(DOCX)

## Acknowledgments

We would like to thank the study participants for sharing their experiences with us. We gratefully acknowledge the contributions of Karl Blanchet to project conceptualisation and protocol development; Zaher Osman (ICRC) and Helmi Mekaoui (DRC) for supporting development of the protocol and tools; Imad El Haddad (MoPH) for project co-ordination; Leila Jaber (LRC) and Christine Bartulec (ICRC) who supported data collection at facility level; Nicole Najjar and Beatrice Tohme who translated transcripts; Sally Yacoub (ICRC) and Michael Bates (DRC) who contributed to data interpretation; and Jytte Roswall and Rima Kighsro Naimi

(DRC) who reviewed versions of the paper; and three peer reviewers whose feedback strengthened the paper. This paper forms part of a broader study undertaken under the 'Partnering for Change' collaboration between the International Committee of the Red Cross (ICRC), Danish Red Cross (DRC) and Novo Nordisk, with the London School of Hygiene and Tropical Medicine acting as independent research partner.

## Author Contributions

**Conceptualization:** Ruth Willis, Chaza Akik, Claudia Truppa, Carla Zmeter, Fabrizio Fleri, Sigiriya Aebischer Perone, Signe Frederiksen, Randa S. Hamadeh, Fouad M. Fouad, Pablo Perel, Bayard Roberts, Éimhín Ansbro.

**Data curation:** Ruth Willis, Chaza Akik, Zeinab El-Dirani.

**Formal analysis:** Ruth Willis, Chaza Akik, Zeinab El-Dirani, Éimhín Ansbro.

**Funding acquisition:** Pablo Perel, Bayard Roberts, Éimhín Ansbro.

**Investigation:** Zeinab El-Dirani.

**Methodology:** Ruth Willis, Chaza Akik, Éimhín Ansbro.

**Project administration:** Ruth Willis, Chaza Akik, Éimhín Ansbro.

**Supervision:** Fouad M. Fouad, Pablo Perel, Bayard Roberts.

**Writing – original draft:** Ruth Willis.

**Writing – review & editing:** Chaza Akik, Zeinab El-Dirani, Claudia Truppa, Carla Zmeter, Fabrizio Fleri, Sigiriya Aebischer Perone, Roberta Paci, Signe Frederiksen, Celine Abi Haidar, Randa S. Hamadeh, Fouad M. Fouad, Pablo Perel, Bayard Roberts, Éimhín Ansbro.

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
