## [Decision Letter · Decision Letter 0]

7 May 2023

PGPH-D-22-01619

Patient experiences of diabetes and hypertension care during an evolving humanitarian crisis in Lebanon: a qualitative study

Dear Dr. Willis,

Thank you for submitting your manuscript to PLOS Global Public Health. After careful consideration, we feel that it has merit but does not fully meet PLOS Global Public Health’s publication criteria as it currently stands. Therefore, we invite you to submit a revised version of the manuscript that addresses the points raised during the review process.

We look forward to receiving your revised manuscript.

Kind regards,

Jianhong Zhou

Staff Editor

Journal Requirements:

2. Please include a complete copy of PLOS’ questionnaire on inclusivity in global research in your revised manuscript. Our policy for research in this area aims to improve transparency in the reporting of research performed outside of researchers’ own country or community. The policy applies to researchers who have travelled to a different country to conduct research, research with Indigenous populations or their lands, and research on cultural artefacts. The questionnaire can also be requested at the journal’s discretion for any other submissions, even if these conditions are not met.  Please find more information on the policy and a link to download a blank copy of the questionnaire here: https://journals.plos.org/globalpublichealth/s/best-practices-in-research-reporting. Please upload a completed version of your questionnaire as Supporting Information when you resubmit your manuscript.”

3. Please send a completed 'Competing Interests' statement, including any COIs declared by your co-authors. If you have no competing interests to declare, please state "The authors have declared that no competing interests exist". Otherwise please declare all competing interests beginning with the statement "I have read the journal's policy and the authors of this manuscript have the following competing interests:"

4. We have noticed that you have uploaded Supporting Information files, but you have not included a list of legends. Please add a full list of legends for your Supporting Information files after the references list.

Additional Editor Comments (if provided):

Reviewers' comments:

Reviewer's Responses to Questions

**Comments to the Author**

1. Does this manuscript meet PLOS Global Public Health’s publication criteria? Is the manuscript technically sound, and do the data support the conclusions? The manuscript must describe methodologically and ethically rigorous research with conclusions that are appropriately drawn based on the data presented.

Reviewer #1: Yes

Reviewer #2: Yes

Reviewer #3: Yes

2. Has the statistical analysis been performed appropriately and rigorously?

Reviewer #1: Yes

Reviewer #2: I don't know

Reviewer #3: N/A

3. Have the authors made all data underlying the findings in their manuscript fully available (please refer to the Data Availability Statement at the start of the manuscript PDF file)?

Reviewer #1: Yes

Reviewer #2: Yes

Reviewer #3: Yes

4. Is the manuscript presented in an intelligible fashion and written in standard English?

Reviewer #1: Yes

Reviewer #2: Yes

Reviewer #3: Yes

5. Review Comments to the Author

Reviewer #1: Very well conducted and written study. This is a very much needed area of research.

I particularly appreciated the typologies described the attention to caregivers and the “workload of demands”.

Minor suggestions

-The six recommendations are very good. I think in addition to grouping them in the conclusions, a table or figure may enhance the impact of the article.

-I suggest that the recommendations include specific ways to bolster support for those who do not have adequate family support as they are clearly more at risk for being in typology 3 category.

-I understand that that another article in this project that will tackle the health system, still, it would be good to add to the last recommendation the importance of strengthening the overburdened public sector and adequate regulation and oversight of the private sector during humanitarian crises in addition to the already stated recommendation of improving the coordination between the humanitarian sectors

Reviewer #2: A joy to read, the background, context and method were clearly explained, the themes congruent with the data. This is a paper that shall influence thinking and action in this crucial shift to patient centred approaches,

Reviewer #3: Dear Dr. Willis,

What a pleasure to review your manuscript titled "Patient experiences of diabetes and hypertension care during an evolving humanitarian crisis in Lebanon: a qualitative study"; very well written, relevant and easy to follow.

I have a few minor points for consideration

- It would be helpful to expand a bit on the conceptual framework used for the "deductive" portion of the qualitative analysis (Page 5, Line 159). This will help identify which findings were driven by prior theory rather than inductively derived from patient experience. This could be done in writing, or maybe visually including the different phases along the continuum of care.

- Was there any member checking (sharing the analyzed data or findings with the participants and asking them to confirm, refute, or provide further insight into the results) following the analyses?

- Finally, while I understand that the transcripts of interviews can't be shared; it would be possible to share some data in terms of how many unique codes were derived from the transcripts, how categories these codes were merged in (you could include those as supplement) and how these categories subsequently led to the identified themes. A table or supplement with categories and themes, in addition to the narrative write-up will give readers the opportunity to review all findings in a single glance.

Congratulations on the important work and excellent paper.

6. PLOS authors have the option to publish the peer review history of their article (what does this mean?). If published, this will include your full peer review and any attached files.

**Do you want your identity to be public for this peer review?** For information about this choice, including consent withdrawal, please see our Privacy Policy.

Reviewer #1: No

Reviewer #2: **Yes: **Marion Lynch

Reviewer #3: **Yes: **Martin Heine

---

## [Decision Letter · Decision Letter 1]

11 Oct 2023

PGPH-D-22-01619R1

Patient experiences of diabetes and hypertension care during an evolving humanitarian crisis in Lebanon: a qualitative study

Dear Dr. Willis,

Thank you for submitting your manuscript to PLOS Global Public Health. After careful consideration, we feel that it has merit but does not fully meet PLOS Global Public Health’s publication criteria as it currently stands. Therefore, we invite you to submit a revised version of the manuscript that addresses the points raised during the review process.

Dear authors,

Thank you for your insightful and well written paper.

I have only one more substantial comment and a couple of minor edits/suggestions.

The first paragraph of the discussion puts significant emphasis on the fact that your analysis did NOT find substantial differences by gender or nationality. Please clearly report this first in the results section. Also, I am not sure if I would emphasize this finding as much – it may be due to the small sample, and it is not the kind of question that I would try to answer with exploratory qualitative research.

Generally, I would recommend beginning the discussion section with the main findings and insights you gained with regards to the study goal/question. I would recommend going into other aspects (like the lack of differences between different groups) after that.

Minor edits and suggestions:

Abstract, line 46: The acronym NCD has not been introduced yet – please either introduce (e.g. in line 30, after spelling out non-communicable diseases) or spell out non-communicable diseases throughout the abstract.

Line 75: burdens? (plural?)

Line 84: patients’?

Line 102: remove comma after unregulated

Please make sure to be consistent in using either US American or British spelling; e.g. privatised vs privatized

Line 291+333 and some of the following headings: I’d suggest removing the colon after the section heading (or alternatively add colons after all sub-headings to be consistent)

Line 379: Consider starting this paragraph with “Patients in this category…”?

Line 422: Verb missing in first half of the sentence (WAS decreasing?), patients’ (plural)

Line 448: I would spell out once more what is meant by “recent events” (the harbour explosion? Covid-19? Inflation?) – with so many things going on all the time, readers who are not completely familiar with the Lebanese context won’t know what is meant here otherwise.

Line 450: “… in  a different publication that focuses on health system aspects/a health system perspective”?

Re: the theme that patients visit multiple providers of care and carry responsibility for bringing with them their medical information – to me, this sounds like a situation that is liable to create duplications and redundancies that, in turn, additionally increase burdens on the patients and the health system (e.g. the multiple performance of a diagnostic procedure, either because the previous results are unavailable/were lost, or because patients are not aware of what procedures they have done before). If you have findings on that, you might want to consider adding them – just a suggestion.

Line 611: Period missing

We look forward to receiving your revised manuscript.

Kind regards,

Nora Gottlieb

Academic Editor

Journal Requirements:

Additional Editor Comments (if provided):

Reviewers' comments:

Reviewer's Responses to Questions

**Comments to the Author**

1. If the authors have adequately addressed your comments raised in a previous round of review and you feel that this manuscript is now acceptable for publication, you may indicate that here to bypass the “Comments to the Author” section, enter your conflict of interest statement in the “Confidential to Editor” section, and submit your "Accept" recommendation.

Reviewer #3: All comments have been addressed

2. Does this manuscript meet PLOS Global Public Health’s publication criteria? Is the manuscript technically sound, and do the data support the conclusions? The manuscript must describe methodologically and ethically rigorous research with conclusions that are appropriately drawn based on the data presented.

Reviewer #3: Yes

3. Has the statistical analysis been performed appropriately and rigorously?

Reviewer #3: Yes

4. Have the authors made all data underlying the findings in their manuscript fully available (please refer to the Data Availability Statement at the start of the manuscript PDF file)?

Reviewer #3: Yes

5. Is the manuscript presented in an intelligible fashion and written in standard English?

Reviewer #3: Yes

6. Review Comments to the Author

Reviewer #3: (No Response)

7. PLOS authors have the option to publish the peer review history of their article (what does this mean?). If published, this will include your full peer review and any attached files.

**Do you want your identity to be public for this peer review?** For information about this choice, including consent withdrawal, please see our Privacy Policy.

Reviewer #3: No

---

## [Editor Report · Decision Letter 2]

20 Oct 2023

Patient experiences of diabetes and hypertension care during an evolving humanitarian crisis in Lebanon: a qualitative study

PGPH-D-22-01619R2

Dear Dr Willis,

We are pleased to inform you that your manuscript 'Patient experiences of diabetes and hypertension care during an evolving humanitarian crisis in Lebanon: a qualitative study' has been provisionally accepted for publication in PLOS Global Public Health.

Best regards,

Nora Gottlieb

Academic Editor